# Atractylenolide II Induces Apoptosis of Prostate Cancer Cells through Regulation of AR and JAK2/STAT3 Signaling Pathways

**DOI:** 10.3390/molecules23123298

**Published:** 2018-12-12

**Authors:** Jing Wang, Moussa Ide Nasser, Salah Adlat, Ming Ming Jiang, Nan Jiang, Li Gao

**Affiliations:** 1College of Veterinary Medicine, Northeast Agricultural University, Harbin 150030, Heilongjiang, China; mingjingw@163.com; 2College of Animal Science, Heilongjiang Agriculture Economics Vocational College, Mudanjiang 157041, Heilongjiang, China; mmingjiang81@163.com; 3Colleges of Life Science and Technology, Dalian University, Dalian Economical Technological Development Zone, Dalian 116622, Liaoning, China; mingjing@dlu.edu.cn (M.I.N.); salahadlat86@gmail.com (S.A.); jiangnan678@sohu.com (N.J.)

**Keywords:** ATR II, JAK2/STAT3, apoptosis, G2/M arrest, AR/PIAS1

## Abstract

Prostate cancer is the most common illness affecting men worldwide. Although much progress has been made in the study of prostate cancer prevention and treatment, less attention has been paid to the molecular mechanism of the disease. The molecular arrangement by which atractylenolide II (ATR II) induces human prostate cancer cytotoxicity was comprehensively examined in the present study. As indicated by the results, ATR II could inhibit prostate cancer cell proliferation and promote DU145 and LNCaP cell apoptosis through induced G2/M cell cycle arrest. The cell apoptosis process induced by ATR II in both DU145 and LNCaP cells was associated with its ability to inhibit androgen receptor (AR) with overexpression of protein inhibitor of activated STAT-1 (PIAS1) and the repression of Janus kinase (Jak2) signaling pathways. The data from the present study demonstrated the antitumor effects and the potential pharmacological application of ATR II as an efficient drug for prostate cancer treatment.

## 1. Introduction

Prostate cancer is the most common form of cancer diagnosed for men. This burden is related to some factors such as genomic, race, age as well as geographic localization [1]. The current treatments for prostate cancer include prostatectomy, chemotherapy, radiotherapy, and immunotherapy [2,3]. These treatments were found to be efficient only at the early stage of prostate cancer development, however many cases where progress to metastasizing occurs, surgery accompanied by chemotherapy has shown promising clinical trials [4]. Prostate cancer is mainly supplemented by numerous molecular vicissitudes, mostly an overexpression of serum interleukin-6 (IL-6) [5].

Plentiful clinical studies reported that in the early stage of prostate cancer, androgen signaling plays a pivotal role in the cells. Nevertheless, over the progression of prostate cancer, an independent androgen state was developed, consequently a chemotherapy molecule which might suppress androgen activity might be promisingly efficient in the prevention and treatment of prostate cancer [6,7]. Moreover, a recent study [8] reported that a hormone depletion might regulate genes such as *Akt* and *ERK1/2* activity in cancer cells to mediate androgen receptor AR down-regulation. *Akt* in return regulates AR and promotes cells inhibition by inducing cells apoptosis and cell cycle arrest [8,9]. Additionally, it was reported that AR transactivation can be activated through IL-6 in human prostate cancer (LNCaP) cells in an androgen-independent manner [10]. Although the phosphoinositide 3-kane (PI3K) pathway is considered the main contributor to IL-6 signaling, its role in the activation of AR by IL-6 is debated. PI3K can activate AR through IL-6. However, activation of AR-mediated by IL-6 does not depend on the PI3K pathway [11].

A potential chemotherapy candidate with such therapeutic property is called atractylenolide II (ATR II) a natural sesquiterpene lactone isolated from *Rhizoma Atractylodis Macrocephalae* (RAM) [12], which have been used as antitumor, and anti-inflammatory agent. ATR II attracts attention, particularly due to its ability to most possess without overt toxicities. Previous studies have been demonstrated that ATR II might induce apoptosis and promote cell cycle arrest in melanoma, gastric, and colorectal cancer through STAT3, nuclear factor kappa-light-chain-enhancer of activated B cells (NF-κB) signaling pathways [12,13,14,15]. However, the mechanism by which ATR II induces prostate cancer cells inhibition remains uncovered. Herein, we purpose to underline the possible in vitro mechanism by which ATR could regulate prostate cancer cells and which indicates ATR II may provide a novel data for the application of ATR II in prostate cancer.

## 2. Results

### 2.1. Cytotoxicity of ATR II on DU145, LNCaP and BPH1

The cytotoxicity of ATR II against human prostate cancer and normal cells was investigated by Cell Titer 96 wells cell proliferation assay. The LNCaP, DU145 as well as BPH1 cells were exposed to increasing concentrations of ATR II 0, 3, 6, 12.5, 25, 50, 100, and 200 µM for 48 h, and 72 h. As shown in Figure 1, ATR II inhibits cells proliferation of DU145 and LNCaP in a dose-time dependent manner, however, no significant effect was observed in the BPH1 cell line. Since the results indicate that the ratio of approximately half minimum concentration IC_50_ was 47, and 94 µM in DU145 and 49, and 100 µM in LNCaP had the highest antiproliferative effect, we therefore chose the two concentrations of 50 and 100 µM to underline the cytotoxicity of ATR II on prostate cancer cell lines.

### 2.2. ATR II Induce Prostate Cancer Cell Cycle Arrest at G2/M Phase Arrest

The regulation of the cell cycle is a key point of cell proliferation, survival, and a disturbance during cell cycle regulation results in cell death by apoptosis [1]. Flow cytometry was performed to determine the stage at which ATR II-induced cell cycle arrest occurs. The results showed an increase in the G2/M phase with the percentage of 6.77 ± 1.6245, 55.755 ± 2.6718 and 64.557 ± 1.6245 in DU145 cells whereas the percentage in LNCaP was 1.457 ± 2.121, 40.531 ± 2.7245 and 62.442 ± 1.70351 respectively after treatment with ATR II 0, 50, and 100 μM for 24 h. G0/G1 and S percentages decreased respectively for the same concentrations (Figure 2A). These results indicate that ATR II-induced LNCaP and DU145 cells cycle arrest in the G2/M phase.

To further confirm the cell cycle arrest in the G2/M phase, western blot was performed to evaluate the G2/M phase regulated proteins. Main studies indicated that G2/M is regulated by cyclin depend kinase 1, cyclin B (CDK1/Cyclin B) complex and downstream p21 protein. As shown in Figure 2B B1/Cdk1 complex. Moreover, it was also reported that human orthologue oncoprotein Mouse double minute 2 (MDM2), a ubiquitin ligase controlling the p53, plays an important role in the regulation of ATR II induce overexpression of the p21 protein, whereas Cyclin B1 and CDK1 proteins expression were inhibited. Taken together, these findings confirmed that the G2/M phase arrest occurred due to down-regulation of Cyclin G2/M [2]. Overexpression of MDM2 resulted in an inactivation of the p53 pathway and inhibits p53-mediated cell cycle arrest. [3]. Herein, ATR II inhibited MDM2 protein expression and overexpressed p53, which may contribute to the cell cycle arrest in the G2/M phase.

### 2.3. ATR II Induces Apoptosis in LNCaP and DU145 Cells through Mitochondrial Intrinsic Pathways

To determine whether the cell cytotoxicity induced by ATR II on prostate cancer cells can induce apoptosis, the cells were labeled with Annexin V and PI and then subjected to flow cytometry. The percentage of cells in the live (Q4), early apoptotic (Q3), late apoptotic (Q2-2), and necrotic (Q1-2) of LNCaP and DU145 cells were determined. The result revealed that the cells death was increased with a dose-dependent manner, and the ratio of apoptotic cells was estimated to be 30% and 70% for DU145 cells while 41% and 72% for LNCaP cells compared to the control. ATR II turned out to induce apoptosis in both DU145 and LNCaP cell line in a dose-dependent manner (Figure 3A).

To further comprehend the molecular mechanism of ATR II-mediated apoptosis in prostate cancer cells, western blot was conducted. Caspases are critical in the regulation of drugs stimulating apoptosis. As illustrated in Figure 3B,C, ATR II can stimulate the activation of p53, which drives an unbalance between *Bcl-2/BAX* complex that consequently induces the release of mitochondrial cytochrome C, which activates Caspases d lead to activated Caspase-3 and consequently cleaved Poly (ADP-ribose) polymerase (PARP), namely the major indicator enzyme of apoptosis. Furthermore, the pre-incubation of DU145 and LNCaP cells with TNF-α induce cells death while caspases inhibitor carbobenzoxy-valyl-alanyl-aspartyl-[O-methyl]- fluoromethylketone (Z-VAD-FMK) eliminated can promote the survival of the cells. These findings promoted the generation of the hypothesis that apoptosis induced by ATR II probably takes place through mitochondrial intrinsic pathways.

### 2.4. ATR II Induce Apoptosis in Prostate Cancer through JAK2/STAT3 Signaling Pathways

To gain insight into the molecular signaling pathways of cell death on how ATR II induces apoptosis in both DU145 and LNCaP cell lines, we first performed western blot to evaluate the expression of the JAK2 protein. Results revealed that ATR II induces phosphorylation of JAK2 in both DU145 and LNCaP cell lines. This inhibition of JAK2 is the main consequence of the activation of STAT3 and its downstream target *ERK*. Herein, we find that ATR II mainly induces the inhibition of STAT3 as well as *ERK* (Figure 4A). To further confirm the apoptosis was induced by ATR II mainly through STAT3 signaling pathways, the STAT3 promoter activity in both DU145 and LNCaP cells line was determined after transient transfecting with the STAT3-Luc vector. As shown in Figure 4B, treatment of prostate cancer cells with ATR II (0, 50 and 100 μM) for 12 h inhibited the STAT3 and promoter activity reached up to 70 and 40% in DU145, whereas LNCaP was reached up to 60 and 12% compared to control. Although the presence of IL-6 restores a part of STAT3 activity, it is clear that ATR II inhibited both IL-6 and STAT3 promoter activities (Figure 4B).

A recent study reported that the activation of JAK2 might promote inhibition of PI3K and its downstream *Akt* signaling [4]. We therefore performed western blot analysis to evaluate whether ATR II could induce these proteins. As shown in Figure 4C, ATR II promotes inhibition of PI3K and *Akt* protein expression.

### 2.5. ATR II Induces LNCaP Cells Apoptosis through Regulation Inhibition of AR and up-Regulation of PIAS1

Researchers have established that an increase of PIAS1 protein expression influences tumor initiation and progression. A recent report suggested that PIAS1 can also act as a co-regulator of AR [5]. We therefore attempt to analyze the expression of these two genes on both LNCaP and Du145 cell line. As shown in Figure 5A, ATR II induces the overexpression of PIAS1 in both DU145 and LNCaP cell lines whereas expression of AR in LNCaP cells was inhibited. However, the expression of AR was null in DU145 cells. To further consolidate the co-regulation of AR by PIAS1, we transfected LNCaP cells with PIAS1 reporter and performed western blot to evaluate the protein expression of AR. As shown in Figure 5B, ATR II induces AR inhibition, while in the transfected cells the expression of AR was restored. To confirm these findings, we transfected LNCaP cells with PIAS1 and treated them with ATR II, then RT-qPCR was conducted to evaluate the expression of AR. As shown in Figure 5B ATR II inhibits expression of AR that was almost restored in the presence of PIAS1. These findings suppose that AR/PIAS1 might have a relationship with the induction of apoptosis mediated by ATR II in prostate cancer.

## 3. Discussion

Through this study, we gathered evidence of the in vitro anti-tumor effects of ATR II on DU145 and LNCaP prostate cancer cells and partially delineated the underlying molecular mechanism. ATR II repressed the transcriptional activity of JAK2/STAT3 and thereby suppressed transcription of oncogenic STAT3 target genes, leading to growth inhibition of DU145 and LNCaP cells. Thus, our findings suggest that ATR II acts at least partly through inhibition of constitutively activated STAT3 and that it represents a novel candidate for prostate cancer chemotherapy drugs. Furthermore, the activation of JAK2 not only included the phosphorylation of STAT3 but also was found to be involved in the activation of downstream mechanisms of cell death activation. We next examined the involvement of PI3K/Akt [6,7]. Notably, *Akt* plays a crucial role in cancer proliferation cell death signaling pathways [8]. *Akt* was reported to promote apoptosis by suppressing AR [9]. Moreover, an increase of Akt in mammalian cells was found to enhance cell proliferation, mainly through inhibition of pro-apoptotic signals, and suppresses some death inducers such as p53 as well.

ATR II was found to induce apoptosis of prostate cancer cells mainly through down-regulation of Akt [10]. Our results in the current study showed that the decreased activity of Akt by ATR II is associated with enhanced expression of p53. Moreover, the expression of Akt and its phosphorylated product indicated a clear dose and time-dependent inhibition of a cell signaling pathway by ATR II.

Additionally, several studies provide that IL-6 overexpression boosts STAT3 DNA binding activity and enhances AR nuclear translocation in LNCaP cells [11]. In this study, we found that ATR II repress IL-6 in both DU145 and LNCaP cell lines through activation of STAT3 pathways.

PIAS1 is a protein inhibitor of activated STAT1 and was a reported co-regulator of AR [5]. PIAS1 is highly expressed in prostate malignancy cancer and the regulation of cell survival as well as cell death [12]. Herein we found that ATR II does not inhibit AR only but enhanced the overexpression of PIAS1 as well. Those findings are in agreement with the previous study on the co-regulation of AR by PIAS1.

The apoptosis induced by ATR II in prostate cancer was also found to be mediated through mitochondrial intrinsic pathways, mainly through mediating an overexpression of tumor suppressor p53 that drive an unbalance on the complex Bcl-2/BAX consequently promote the release of cytochrome c, and therefore by cascades activation of caspase9/3.

The cell proliferation is mainly controlled by cell cycle arrest, which is controlled by several checkpoints [13]. G2/M phase in particular is regulated by the complex CDK1/cyclin B [14]. ATR II causes DU145 and LNCaP cell cycle arrest at the G2/M phase mainly through overexpression of p21 protein that consequently drives the inhibition of CDK1/cyclin B complex, thereby leading to cells cycle arrest in the G2/M phase.

In summary, the present study predominantly explained the mitochondrial anti-proliferation role and pro-apoptosis role of ATR II in DU145 and LNCaP cells (Figure 6). ATR II was identified to induce apoptosis through mitochondrial intrinsic pathways. The apoptosis induced by ATR II was also associated with its ability to inhibit AR with overexpression of PIAS1. Furthermore, ATR II drives phosphorylation of JAK2 that results in the activation of PI3K and its downstream Akt protein, as well as inhibition of STAT3 and its downstream target *ERK.* Finally, ATR II-induced LNCaP and DU145 cytotoxicity was associated with G2/M phase cell cycle arrest.

## 4. Materials and Methods

### 4.1. Chemicals and Reagents

ATR II was purchased from Baoji Herbest Bio-tech Co., Ltd. (Beijing, China). Fetal bovine serum (FBS) was purchased from Hangzhou Sijiqing Biological Engineering Material, Co., Ltd. (Beijing, China). Dulbecco’s modified Eagles medium (DMEM) was purchased from Gibco (Thermo Fisher Scientific, Inc., Waltham, MA, USA)

All ethical and laboratory guidelines were followed and no misidentified cell lines were used. The LNCaP, DU145, BPH1 cell line was purchased from ATCC (Shanghai, China).

### 4.2. Cell Culture

The LNCaP, DU145, and BPH1 cell line was cultured and maintained in DMEM medium containing 10% of fetal bovine serum (FBS) (Gibco, Beijing, China). Cells were incubated at 37 °C in a humidified atmosphere with 5% CO_2_ and allowed to grow to 70–80% confluence.

### 4.3. Cell Proliferation Assay

Cells were cultured and maintained in DMEM containing 1% FBS, incubated at 37 °C in a humidified atmosphere of 5% CO_2_ and allowed to grow to 70–80% confluence. Afterward, the cells were harvested and seeded in a 96-well dish to a final concentration of 5 × 10^3^ cells/well and incubated in DMEM medium containing 1% FBS for 24 h. Then the cells were treated with different concentrations of ATR II (0, 6, 12.5, 25, 50, 100, and 200 μM) and incubated for 48 or 72 h. Following this, 20 µL of 3-(4,5-dimethylthiazol-2-yl)-2,5-diphenyltetrazolium bromide(MTT) solution (5 mg/mL) was added to each well and incubated for 4 h. Finally, the medium was discarded and 150 µL of DMSO was added to each well. The plates were read at a wavelength of 570 nm using Varioskan Flash Multimode Reader (Thermo Fisher Scientific, Tokyo, Japan). A total of 6 reduplicate wells were used for each treatment, and experiments were repeated 3 times and their inhibition ratio (I%) based on the equation:I % = [A570 (control) − A570 (treated)]/A570 (control) × 100 where I is the inhibition rate and A is the absorbance at 570 nm.

### 4.4. Annexin V/PI Assay for Apoptosis

Apoptotic cells were investigated by Annexin V/PI staining followed by flow cytometry according to the manufacturer’s protocol. LNCaP and DU145 cells (5 × 103 cells/well) were cultured in 6 wells plates and treated with different concentrations (0, 50 and 100 μM) of ATR II for 24 h. The cells were then washed twice with PBS and stained with 5 µL of Annexin V-FITC and 10 µL of PI in 500 µL binding buffer for 15 min at room temperature in the dark. The apoptotic cells were determined by flow cytometry (Cytomics FC 500; Beckman Coulter Inc., Miami, FL, USA).

### 4.5. Determination of Cells Cycle Distribution

The cell cycle distribution in different phases following exposure to ATR II was analyzed by flow cytometry. In brief, LNCaP and DU145 cells (5 × 10^3^ cells/well plates) were seeded into 6-well plates and exposed to different concentrations (0, 50 and 100 μM) of ATR II for 48 h. The cells were harvested, washed with PBS twice and subsequently fixed with 70% ethanol for 2 h. The cells were centrifuged and washed with PBS, resuspended in 500 µL of buffer containing 10 µL of RNase and 25 µL of PI, and incubated at room temperature in the dark for 15 minutes. The distributions of the cell cycle were determined by flow cytometry (Cytomics FC 500; Beckman Coulter, CA, USA).

### 4.6. Western Blot Analysis

The protein expression regulated by ATR II was analyzed by western blotting and following the protocol as previously described with small modifications. In brief, LNCaP and DU145 cells were treated with ATR II (0, 50 and 100 µM) for 24 h then harvested and lysed with Radioimmunoprecipitation assay buffer (RIPA buffer). Afterward, the insoluble protein lysate was removed by centrifugation at 12,225 × *g* for 15 min at 4 °C. The protein concentrations were determined using a NanoDrop 1000 spectrophotometer (Thermo Fisher Scientific, Inc., Wilmington, DE, USA). An equal amount of protein was loaded on SDS-PAGE electrophoresis gel (10 or 12% according to the protein size) and the gel transferred onto polyvinylidene fluoride membrane (PVDF). The membrane was blocked in 5% (w/v) non-fat milk and incubated for 2 h. The membrane was thereafter incubated with appropriate primary antibodies phospho-PI3 kinase, phospho-*Akt* (Ser473), *Akt*, JAK-2, p-Jak-2, *p-ERK*, STAT3, p-STAT3, *Bcl-2 BAX* (1:1000) Cell Signaling Technology, Inc. (Danvers, MA, USA), Cyclin E, CDK1, p21, mouse, anti-rabbit, p53, MDM2, AR, PIAS1 (1:200) Santa Cruz Biotechnology, Inc. (Dallas, TX, USA) at 4 °C overnight and washed three times with a Tris-buffered saline-Tween solution (TBST). Finally, the blots were incubated with appropriate secondary antibodies (anti-rabbit or anti-mouse horseradish peroxidase conjugated) for 1 h at room temperature, then washed with TBST for 30 min. Signals were detected using ECL plus chemiluminescence kit on X-ray film (EMD Millipore, Billerica, MA, USA).

### 4.7. Reverse Transcription-Quantitative Polymerase Chain Reaction (RT-qPCR)

Total RNA was extracted using TRIzol (Life Technologies, Thermo Fisher Scientific, Shanghai, China), according to the manufacturer’s protocols. Briefly, first-strand cDNA was reverse-transcribed from 1 μg total RNA using the Super-Script First-Strand cDNA System (Invitrogen, Thermo Fisher Scientific, Shanghai, China) and amplified by Platinum SYBR Green qPCR Super Mix-UDG (Invitrogen, Thermo Fisher Scientific, Shanghai, China). A master mix was prepared for each PCR reaction, which included Platinum SYBR Green qPCR Super Mix-UDG, forward primer, reverse primer and 10 ng of template cDNA. PCR conditions were 10 min at 95 °C, followed by 50 cycles at 95 °C for 30 s and 60 °C for 1 min and 72 °C for 30 s. The forward and reverse primer sequences for AR were 5′-AAGCCAGAGCTGTGCAGATGA (forward) and 5ʹ-TGTCCTGCAGCCACTGGTTC-3ʹ (reverse), the forward and reverse primer for β-actin were 5ʹ-CACGATGGAGGGGCCGGACTCATC-3ʹ (forward) and 5ʹ-TAAAGACCTCTATGCCAACACAGT-3′ (reverse). Relative gene expression was obtained after normalization with β-actin and determination of the difference in threshold cycle (Ct) between treated and untreated cells was performed using the 2^−ΔΔCq^ method [15].

### 4.8. Transient Transfection and Luciferase Assay

Luciferase assays were performed as reported previously [16]. Briefly, transient transfections were performed using Lipofectamine 2000 (Invitrogen, Thermo Fisher Scientific, Shanghai, China) following the manufacturer’s protocol. Cells were seeded into 48-well plates for 16 h and transfected with each of reporter plasmids 100 ng or with 25 ng of expression plasmid in the presence of Renilla Luciferase control pREP7 vector 25 ng, and then treated with ATR II 0, 50 and 100 μM. Firefly Luciferase activities were calculated by using the Dual-Luciferase reporter assay system (Promega Corporation, Madison, WI, USA) and the ratio of Firefly Luciferase activity to Retile Luciferase activity was measured as relative Luciferase activity.

### 4.9. Statistical Analysis

All statistical analyses were performed using Origin Lab software version 8.0 (Origin Lab, Northampton, MA, USA). The statistically significant differences between groups were determined by two-tailed paired Student’s *t*-test. *p* < 0.05 was considered statistically significant.

## Figures and Tables

**Figure 1 molecules-23-03298-f001:**
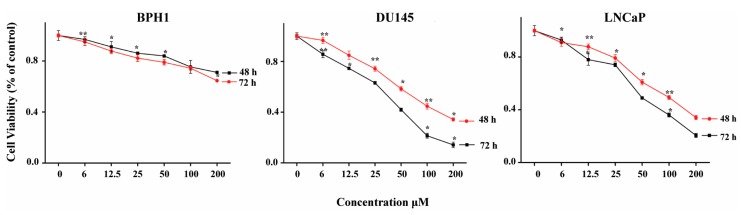
Cytotoxicity of atractylenolide II (ATR II) on prostate cancer and normal cells line. DU145 and LNCaP prostate cancer cells line, as well as BPH1 prostate normal cells line, were treated with increasing concentrations 0, 6, 12.5, 25, 50, 100, and 200 µM of ATR II, Proliferation was assessed after 48 h and 72 h, as described in Materials and Methods. Each bar represents the mean ± standard deviation of three experiments. * *p* < 0.05; and ** *p* < 0.01 compared with the control.

**Figure 2 molecules-23-03298-f002:**
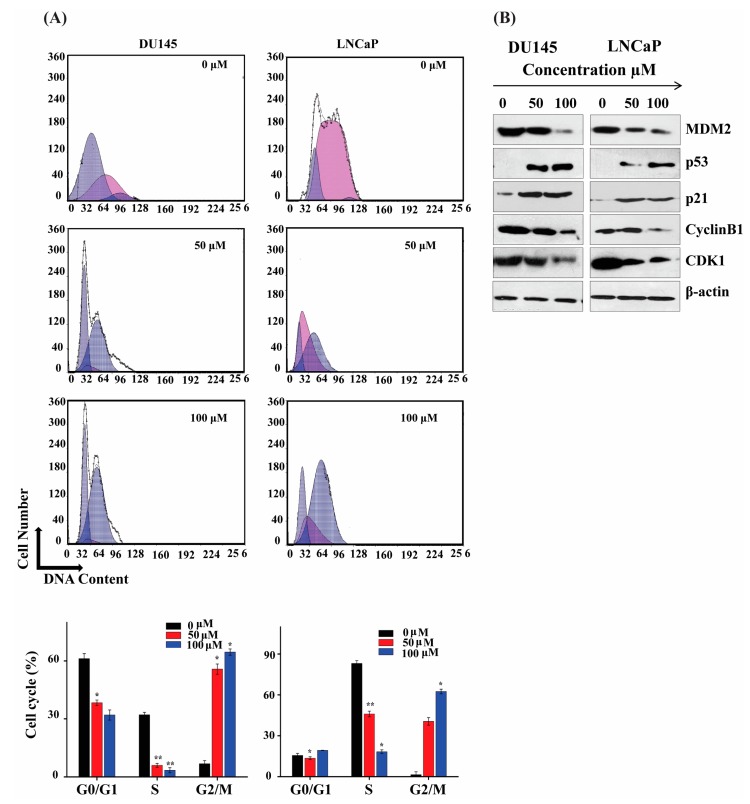
ATR II induce prostate cancer cell cycle arrest at the G2/M phase. (**A**) DU145 and LNCaP cells were treated with ATR II (0, 50 and 100 μM) for 48 h, and stained with PI for flow cytometric analysis. Histograms show a number of cells/channel (y-axis) vs. DNA content (x-axis). The values indicate the percentage of cells in the indicated phases of the cell cycle. The data are shown the representative of three independent experiments with similar results. * *p* < 0.05; and ** *p* < 0.01 compared with the control. (**B**) DU145 and LNCaP cells were treated with ATR II (0, 50 and 100 μM) for 48 h, then cellular proteins were extracted and the level of protein 53 (p53), p21 cyclin B1, cyclin depend kinase 1 (CDK1), Mouse double minute 2 (MDM2) as well as β-actin (control) were detected by western blotting.

**Figure 3 molecules-23-03298-f003:**
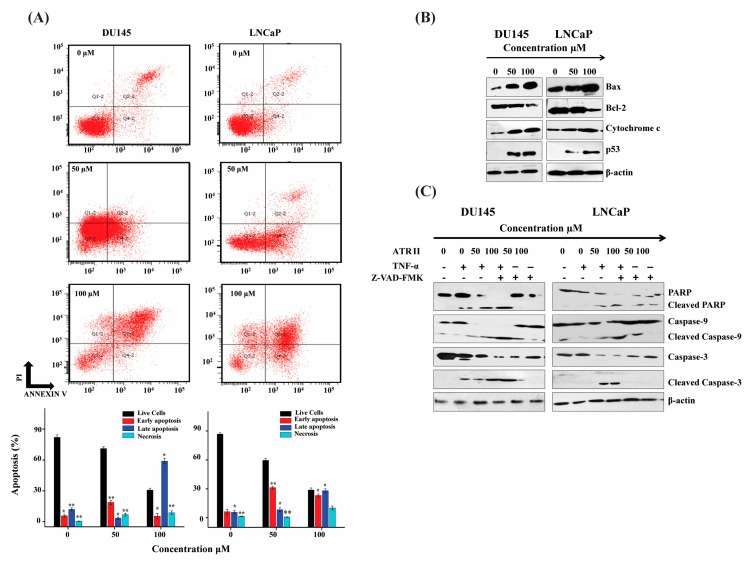
ATR II induces apoptosis in prostate cancer cells line through mitochondrial intrinsic pathways. (**A**) DU145 and LNCaP cells were treated, with or without ATR II (0, 50 and 100 μM for 48 h) then stained with annexin V/PI, the lower right quadrant showed V^+^/PI^−^ annexin, the upper right quadrant showed V^+^/PI^+^ annexin cells. Each bar represented the percentage (mean ± SD of triplicate determinations) of both annexin V^+^/PI^−^ and V^+^/PI annexin cells (* *p* < 0.05; and ** *p* < 0.01 compared with the control). (**B**) DU145 and LNCaP cells were treated with (0, 50 and 100 μM for 48 h) ATR II then the expression of *BAX*, B-cell lymphoma 2 (*Bcl-2*), p53, cytochrome C as well as β-actin were detected by western blot. (**C**) Cells were pre-incubated with 20 μM carbobenzoxy-valyl-alanyl-aspartyl-[O-methyl]-fluoromethylketone (Z-VAD-FMK) and tumor necrosis factor alpha (TNF-α) for 30 min and subsequently treated with or without ATR II 0, 50 and 100 μM for 48 h. The cellular proteins were extracted to detect the levels of PARP, Caspase-9, and Caspase-3 as well as β-actin as control by western blotting.

**Figure 4 molecules-23-03298-f004:**
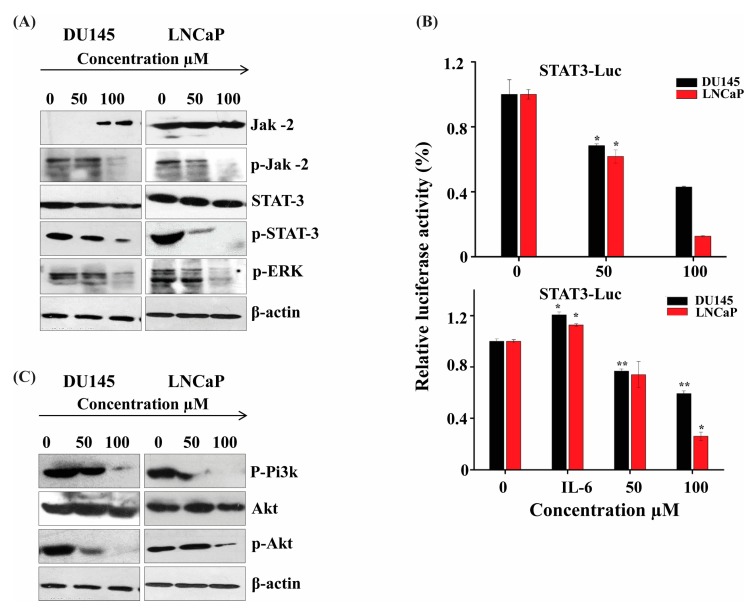
ATR II induces apoptosis in prostate cancer through JAK2/STAT3 signaling pathways. (**A**) DU145 and LNCaP cells were pre-incubated with ATR II at different concentrations (0, 50 and 100 μM) for 48 h, then the protein expression of JAK2, STAT3, and *ERK* was detected by immunoblotting. (**B**) ATR II inhibits constitutive STAT3 and IL-6 in DU145 and LNCaP cells line: cells were transiently transfected with the M67-Luciferase plasmid. After 16 h of transfections, cells were treated with the ATR II (0, 50 and 100 μM) for 12 h and then harvested for Luciferase assays. Cells were treated with IL-6 10ng/mL for stimulation. (* *p* < 0.05; and ** *p* < 0.01 compared with the control). (**C**) DU145 and LNCaP cells were treated with or ATR II (0, 50 and 100 μM) for 48 h. The cellular proteins were extracted to detect the levels of Akt, p-Akt p-Pi3k as well as β-actin (control) by western blotting.

**Figure 5 molecules-23-03298-f005:**
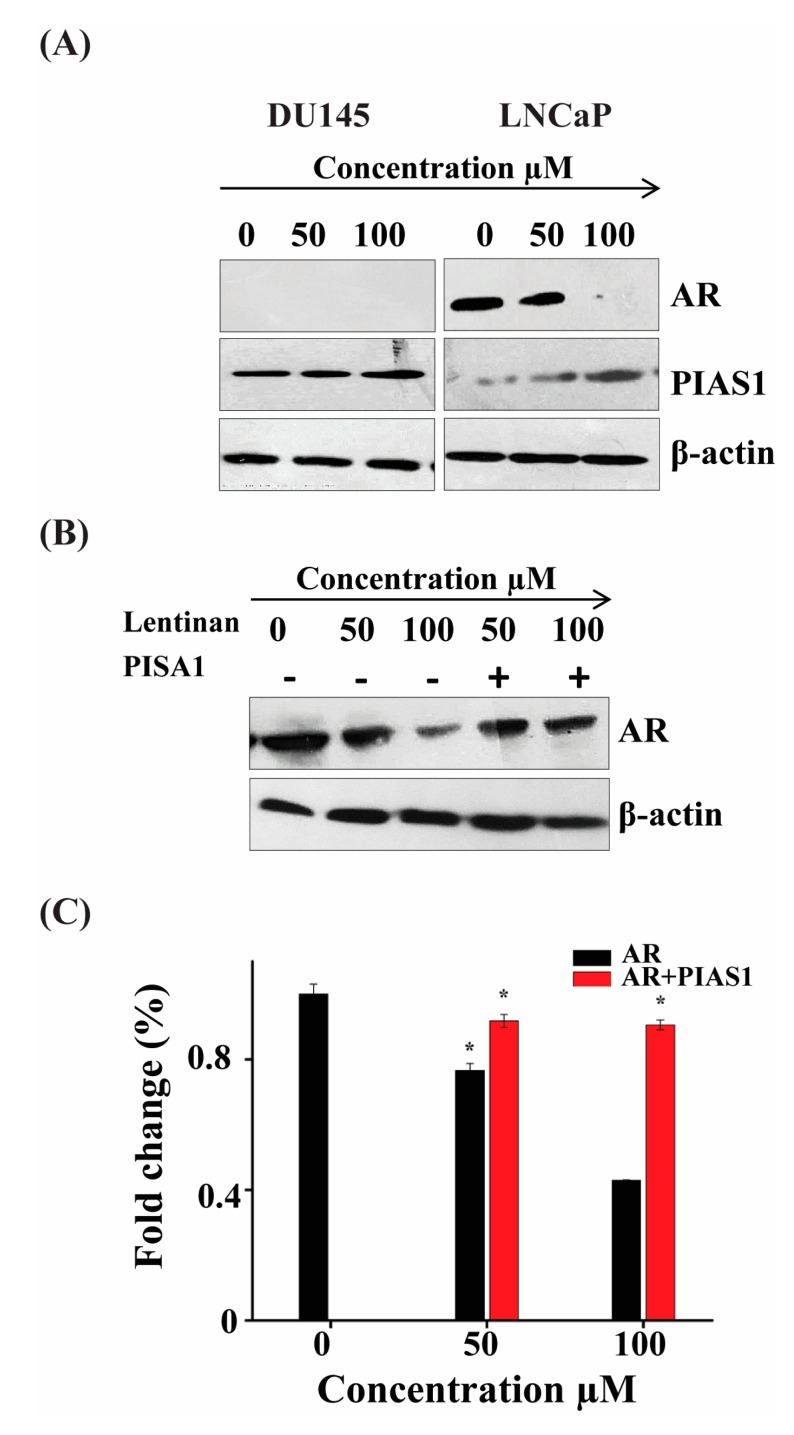
ATR II induces LNCaP cells apoptosis through regulation inhibition of AR and up-regulation of PIAS1. (**A**) LNCaP cells were treated with ATR II (0, 50 and 100 μM) for 48 h. The cellular proteins were extracted to detect the levels of AR and PIAS1 as well as β-actin (control) by Western blotting. (**B**) LNCaP cells were transfected with or without PIAS1, then treated by ATR II (0, 50 and 100 μM) for 48 h then the protein expression of AR as well as beta-actin were evaluated by western blot. (**C**) LNCaP cells line were transfected with or without PIAS1, then treated by ATR II 0, 50 and 100 μM for 48 h then RT-PCR was performed to detect the fold change of AR. (* *p* < 0.05 compared with the control).

**Figure 6 molecules-23-03298-f006:**
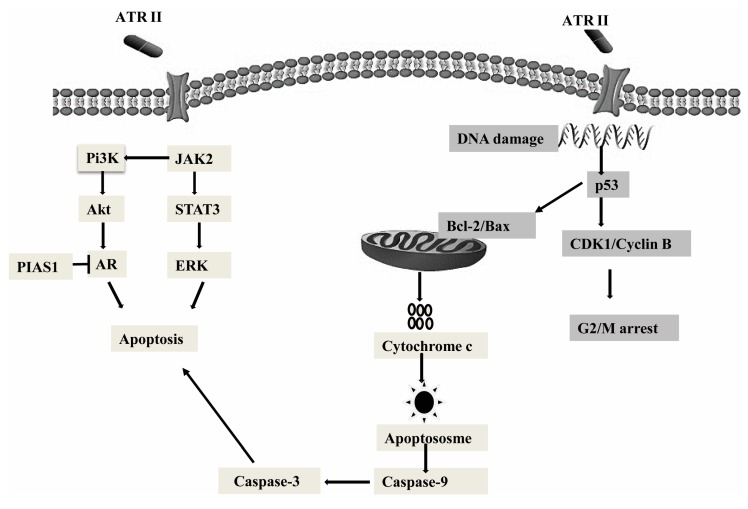
Hypothetical mechanism action of cancer cells. ATR II induces prostate cancer cells apoptosis through inhibition of JAK2/STAT3 signaling that promotes the inhibition of Pi3K/Akt as well as AR with overexpression of PIAS1, those were mainly drive through mitochondrial intrinsic pathways. Moreover, ATR II might induce prostate cancer DNA damage with conduct to cell cycle arrest at G2/M phase.

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
