# Peer review of "Atractylenolide II Induces Apoptosis of Prostate Cancer Cells through Regulation of AR and JAK2/STAT3 Signaling Pathways"

_molecules, 2018, doi:10.3390/molecules23123298_

Reviewer 1 Report

In the manuscript entitled Atractylenolide II Promote Prostate Cancer Cells Cytotoxicity by Mediated Apoptosis through Regulation of AR and JAK2/STAT3 Signaling Pathways” the molecular arrangement by which atractylenolide II (ATR II) induces human prostate cancer cytotoxicity was comprehensively examined. ATR II could inhibit prostate cancer cell proliferation and promote DU145 and LNCaP cell apoptosis through induced G2/M cell arrest. The cell apoptosis process induced by ATR II in both DU145 and LNCaP cells was associated with its ability to inhibit androgen receptor with overexpression of PIAS1 and the inhibition of STA3/JAK2 signaling pathways. The data from the present study demonstrated the antitumor effects and the potential pharmacological application of ATR II as an efficient drug for prostate cancer treatment.

Specific Comments:

This is technically well performed study but the authors need to address several missing links before it can be even considered for publication. Specific points that the authors need to address are as follows:

The molecular mechanism(s) by which ATRII exhibits its anticancer  effects are not clear? For  example, whether deletion of STAT3 by siRNA abrogates the observed anticancer effects of ATRII should be analyzed?

The effect of ATRII should also be analyzed on normal prostate epithelial cells to rule out potential cytotoxicity. Also, acute toxicity studies should be performed to establish the safety of the compound.

Also, the anti-invasive/anti-migratory effects of ATRII should  be analyzed.

A limited in vivo study in appropriate xenograft/orthotopic mouse model will greatly increase the impact of the reported in vitro findings.

Several typographical errors were noted throughout the manuscript and also even in the abstract.

Author Response

Dear Reviewer

Thank you for your comment and reviewer our manuscript.

In the manuscript entitled Atractylenolide II Promote Prostate Cancer Cells Cytotoxicity by Mediated Apoptosis through Regulation of AR and JAK2/STAT3 Signaling Pathways” the molecular arrangement by which atractylenolide II (ATR II) induces human prostate cancer cytotoxicity was comprehensively examined. ATR II could inhibit prostate cancer cell proliferation and promote DU145 and LNCaP cell apoptosis through induced G2/M cell arrest. The cell apoptosis process induced by ATR II in both DU145 and LNCaP cells was associated with its ability to inhibit androgen receptor with overexpression of PIAS1 and the inhibition of STA3/JAK2 signaling pathways. The data from the present study demonstrated the antitumor effects and the potential pharmacological application of ATR II as an efficient drug for prostate cancer treatment.

Specific Comments:

This is technically well performed study but the authors need to address several missing links before it can be even considered for publication. Specific points that the authors need to address are as follows:

The molecular mechanism(s) by which ATRII exhibits its anticancer effects are not clear? For example, whether deletion of STAT3 by siRNA abrogates the observed anticancer effects of ATRII should be analyzed? The effect of ATRII should also be analyzed on normal prostate epithelial cells to rule out potential cytotoxicity. Also, acute toxicity studies should be performed to establish the safety of the compound.

Thank you for your comment, we have observed that ATR II doesn’t have any significant effect on BPH1(Normal Prostate cell line), therefore we didn’t do further experiment on this cells line. MTT results also showed that we can use ATR II safely since it can only significantly effect on prostate cancer cell line.

Also, the anti-invasive/anti-migratory effects of ATRII should be analyzed. A limited in vivo study in appropriate xenograft/orthotopic mouse model will greatly increase the impact of the reported in vitro findings.

Thank you for your comment, we will performed the migratory as well as in vivo effect of ATR II in ours further study.

Several typographical errors were noted throughout the manuscript and also even in the abstract.

We appreciate reviewer’s time and valuable comments very much. We noticed that too. We have read, made revision and corrected the errors.

Many thanks

Reviewer 2 Report

Dear Editor,

I reviewed the manuscript by  Jing  et al., entiteled ‘’  Atractylenolide II Promote Prostate Cancer Cells  Cytotoxicity by Mediated Apoptosis through  Regulation of AR and JAK2/STAT3 Signaling Pathways   ‘’

The study is interesting. And well written. However, the quality of the presented data is very bad. I do not know if the presented data are reproducible. For example the analysis of apoptosis by annexin V/PI. The gate is not adjusted.  Atractylenolide II –induced apoptosis is very strong. Why the authors does not reduce the concentration and the time to show minima concentration and minimal time period that may 20% to 30% of cell death via apoptotic mechanism. Also Western blot analyisi is very bad need to be replaced by a new one.  Also the title is not accurate. It is better to say: Atractylenolide II  induces apoptosis of …..instead off to promotes…….

Also, the labelling of the figures makes the reader confuse. All Western blots need to be repeated.

Many thanks

Author Response

Dear Reviewer

Thank you for your comment and reviewer our manuscript.

The study is interesting. And well written. However, the quality of the presented data is very bad. I do not know if the presented data are reproducible. For example the analysis of apoptosis by annexin V/PI. The gate is not adjusted.  Atractylenolide II –induced apoptosis is very strong. Why the authors does not reduce the concentration and the time to show minima concentration and minimal time period that may 20% to 30% of cell death via apoptotic mechanism.

Thank you for your comment, we have use the IC50 and half of IC50 value, however we will take note of your precious comment, we have also adjusted gate, we will decreased the concentration of ATR II in our further study, for indeed investigation.

 Also Western blot analysis is very bad need to be replaced by a new one. 

Thank you for this observation. Actually, we have performed western blot manually, the reason why are the bands not clear and it’s quite difficult to do it again meantime as our lab doesn’t have any other machine to do it. 

Also the title is not accurate.

It is better to say: Atractylenolide II induces apoptosis of …..Instead off to promotes…….

Thank you for this observation, we have change the title by “Atractylenolide II induce apoptosis of prostate Cancer Cells through Regulation of AR and JAK2/STAT3 Signaling Pathways”

Also, the labelling of the figures makes the reader confuse. All Western blots need to be repeated.

Yes, we agree. We have made proper changes. Thanks for the comment. Regarding western blot it’s quite hard for us to repeat all the western blot as we are using western blot manually, which make it hard to finish it within the requested time, thanks for understanding.

Many thanks

Round  2

Reviewer 1 Report

The authors have addressed all my concerns.

Author Response

No comment

Reviewer 2 Report

Dear editor,

I Feel für Authors Ads Not interesting to Adresse für Comment of für reviewers. Accordingly, für manuscript can not ne Publisher in the present form. Many thanks

Author Response

Dear Reviewer

Thank you for your comment and reviewer our manuscript.

The study is interesting. And well written. However, the quality of the presented data is very bad. I do not know if the presented data are reproducible. For example the analysis of apoptosis by annexin V/PI. The gate is not adjusted.  Atractylenolide II-induced apoptosis is very strong. Why the authors do not reduce the concentration and the time to show minima concentration and minimal time period that may 20% to 30% of cell death via an apoptotic mechanism.

Thank you for your comment, we have to use the I

C50 and half of the IC50 value, however, we will take note of your valuable comment, we have also adjusted gate, we will decrease the concentration of ATR II in our further study, for in vivo investigation.

 Also, Western blot analysis is very bad need to be replaced by a new one. 

Thank you for this observation. Actually, we have performed western blot manually, the reason why are the bands not clear and it’s quite difficult to do it again meantime as our lab doesn’t have any other machine to do it. We have to repeat the western blot without getting better results. Hope so you might understand this issue.

Also, the title is not accurate.

It is better to say: Atractylenolide II induces apoptosis of …..Instead off to promotes…….

Thank you for this observation, we have to change the title by “Atractylenolide II induces apoptosis of Prostate Cancer Cells through Regulation of AR and JAK2/STAT3 Signaling Pathways”

Also, the labeling of the figures makes the reader confused. All Western blots need to be repeated.

Yes, we agree. We have made proper changes. Thanks for the comment. Regarding western blot, it’s quite hard for us to repeat all the western blot as we are using western blot manually, which make it hard to finish it within the requested time, thanks for understanding.

Many thanks